# Asparagus (*Asparagus officinalis* L.) Root Distribution Varies with Cultivar during Early Establishment Years

**Daniel Drost**

Department of Plants, Soils and Climate, Utah State University, 4820 Old Main Hill, Logan, UT 84322, USA; dan.drost@usu.edu

**Abstract:** Soil type, crop management practices, annual plant growth patterns, and seasonal changes have all been shown to influence asparagus (*Asparagus officinalis*) roots. This study describes the changes in root growth that occur over the three establishment seasons of three asparagus cultivars. Starting one year after planting, asparagus root length density and biomass were estimated from soil cores (55 mm diam. × 0.2 m long) collected to 0.9 m at three locations adjacent to the row (0.15, 0.3, and 0.6 m from row center). Samples were collected each spring during the spear harvest period (late April to early May). Soil cores were divided into 0.15 m lengths and fleshy roots were collected for the soil, root length density determined, and dry weights measured. The year of sampling had a significant effect on root development and, as time progressed, fleshy asparagus roots grew deeper into the soil. There was no difference in fleshy root length or fresh weight between the three cultivars evaluated. However, root distribution patterns varied between the cultivars. Fleshy roots for the cultivars Atlas and Jersey Giant extended further from the crown and deeper in the soil when compared to Guelph Millennium. Results can be used to improve crop management practices and increase our understanding of the dynamic changes of root development that occur over time in asparagus.

**Keywords:** root sampling; rooting depth; Atlas; Guelph Millennium; Jersey Giant

## 1. Introduction

For those who grow asparagus (*Asparagus officinalis* L.), the growth of spears and the vigor of fern are commonly used to assess present and future crop performance [1–5]. Determining yield by assessing aboveground growth often leads to over-estimations of productivity since plant performance varies widely among locations, cultivars, level of plant care, and plant age, and in many cases, the causes of variations cannot be identified. High productivity depends on maintaining healthy ferns throughout the summer, with the ability to balance root recharge and crown extension during the summer [5,6]. Thus, growing a large crown, developing lots of storage roots, and producing many buds are the keys to high productivity [6]. The level of stored carbohydrates (CHOs) in the root system is one good indicator of yield potential [7–14]. Wilson et al. [14] compared the root system to an "engine" and stored CHO as the "fuel" that drives crop performance. Since root CHOs are a key determinant of productivity [8,10,11,14], a clear idea of root size can be used to assess crop performance [15–20].

The belowground parts of asparagus are known collectively as the crown and consists of a branching rhizome and numerous fleshy storage roots [5,6]. The fleshy storage roots are large diameter (>2 mm) adventitious roots, are initiated off of the rhizome [15–17], and serve as nutrient and carbohydrate storage organs [8,11,14,15]. Fleshy roots grow and elongate for several years [7,17,19] and often extend outward and downward in the soil profile for several meters [18–20]. Many things affect asparagus root growth, including soil type [18,21], soil tillage [22–24], water management [25–28], plant nutrition [6,29–31], and cultivars [10,20,26].

Changes in asparagus root growth during the first year after plant establishment are well documented [9,32,33], as are root growth patterns in mature plantings over one complete growing season [21,24,25]. Seasonal changes in root distribution patterns have also been documented [18,24,25] but longer-term studies on the dynamic changes in rooting patterns over many seasons are not fully understood. Understanding the dynamics of root growth is important since farm management practices that encourage root development are required to optimize plant productivity [34–36]. Research clearly demonstrated that root damage associated with asparagus tillage has a major impact on stand longevity and productivity [22–24] and increases the susceptibility of plants to crown and root rot diseases [37,38]. In addition, soil tillage causes soil compaction, which restricts root development [39–42], compromises the plants access to water and nutrients [27,28], and increases the susceptibility to disease [37] and pests [35],all of which have direct impacts on yield, quality, and production costs. Interrow application of compost or straw mulch in combination with shallow soil disturbance is an effective approach to mitigate deep compaction in asparagus interrows associated with annual tillage, harvest-related foot traffic, and other associated harvest operations [39]. Tillage operations, such as the sub-soiling of interrows, can pose a high risk of damage to asparagus root systems [40,41], and asparagus varieties differ in their susceptibility to tillage related damage [42]. These differences were attributed to different patterns of root distribution within the planted row and interrow areas of the field. A better understanding of the dynamics of root change would help farmers, farm managers, and researchers optimize practices that ensure high productivity is achieved and maintained.

Understanding asparagus root growth can also help with nutrient and water management. Mature asparagus plantings are only marginally responsive to added fertilizers [29–31,35]. The lack of response is due in part to asparagus's extensive perennial root system [6]. Understanding root biomass changes provided more information regarding productivity [21,24] and helped explain productivity differences that fern-related data could not explain [1–3]. Since asparagus roots contain high levels of nutrients, this buffers the plants response to applied nutrients. Asparagus also effectively acquires and recycles nutrients and has relatively low nutrient removal levels during harvest [29–31]. However, further work on long-term changes in root growth are needed to better understand how quickly the root system develops and how crop management practices impact asparagus growth and productivity.

In asparagus, studies have noted variation in cultivar response to water stress [25–28,43,44], but the basis for this variation is often unclear. Schaller and Paschold [44] reported that cultivar Grolim had more sensitive stomata when exposed to water deficits than Gijnlim, thus greater drought tolerance. However, differences in rooting characteristics were not adequately assessed. Brainard et al. [26] noted that variation in cultivar response to irrigation appeared to be related to differences in rooting depth. While direct root distribution was not measured [26], they noted that under drought conditions, soil water was extracted from deeper in the soil profile for cultivar Jersey Supreme compared to Guelph Millennium suggesting differences in root distribution. Water use differences may be related to soil compaction issues as asparagus beds vary in penetration resistances, which can influence water infiltration [39–42]. Based on these findings, differences among cultivars seem to suggest that there is significant root size differences and that these differences might be correlated with the yield potential of mature plants [3]. There is limited information on varietal differences in rooting patterns in asparagus. Therefore, the objectives of this study was to evaluate the changes in fleshy root growth and distribution over the early establishment years of three different asparagus cultivars.

## 2. Materials and Methods

The study was conducted at the Utah State University Greenville Research Farm (lat. 41.766° N, long. 111.811° W, 1382 m elevation, 135 freeze-free days, average last frost: 15 May, and USDA hardiness zone 5), Logan, Utah, from 2016 to 2018. Seasonal tem-

peratures vary from an average low of −4.1 °C in January to an average high of 22.8 °C in July (Utah Climate Center; www.climate.usu.edu, (accessed 4 November 2022)) with a mean annual precipitation of 505 mm (mostly as snow). The soil was a Millville silt loam (coarse silty, carbonatic, mesic typic haploxeroll) that holds approximately 225 mm of water at field capacity in a 1.5-m profile (NRCS). Soil test results, taken prior to planting, indicated pH 7.3, with 10.6 mg/kg of P, 129 mg/kg of K, organic matter 1.8%, and salinity of 1.35 dS/m. Prior to planting, the site received a broadcast application of KCl (100 kg/ha of K), which was rototilled into the soil, and furrows (20 cm deep) were formed in preparation for planting. Concentrated superphosphate (125 kg/ha of P) was scattered in the furrow bottom, and a shank was used to incorporate the fertilizer and break up the compacted layer formed during furrow formation.

For the study, 90-day-old seedling transplants of asparagus cultivars, Atlas F1 (Walker Brothers Inc., Pittsgrove, NJ, USA; https://walkerseed.com/ (accessed 12 December 2022)), Guelph Millennium' (Fox Seeds, Simcoe, ON, Canada; https://foxseeds.com/ (accessed 12 December 2022)), and Jersey Giant (cultivar discontinued) were grown in a heated greenhouse (20°/16 °C day/night). Atlas is a vigorous, high yielding, early season, large spear cultivar adapted to hot growing conditions. It has good disease tolerance but is not adapted to very cold production regions. Guelph Millennium is a vigorous, high yielding, later season, medium spear cultivar adapted to cool to cold growing conditions. It has good disease tolerance, is adapted to different soil types, and tolerates cold winters. Jersey Giant is a vigorous, high yielding, early season, large spear cultivar that was the standard for local production. It has good disease tolerance, is widely adapted, and tolerates cold winters. Seed were sown in 50-cell seedling trays filled with a 1:1:1 ratio of peat, perlite, and vermiculite. After emergence, plants were watered daily and fed twice weekly with a 100 mg/L$^{-1}$ of 20N:20P:20K soluble fertilizer. Seedling asparagus plants at planting had four to five fern, seven to ten storage roots, and a crown (roots + rhizome) dry weigh of 12.0 g (±2.9).

Seedlings were hand planted on 16 April 2015 in 1.5 m wide rows with an in-row plant spacing of 30 cm (22,222 plants/ha). Rows were 6.1 m long, with two rows per cultivar, with a guard row separating each cultivar. There were four replications and replicates were separated from each other by 1.5 m. During the establishment year, rows were hand weeded and between-row areas periodically rototilled to gradually fill in the furrows. In subsequent years, the site was managed no-till and weeds were controlled with herbicides (glyphosate, 1.68 kg/ha a.i. and pendimethalin, 4.67 L//a a.i.) applied prior to spear emergence in spring and again after the final harvest. Nitrogen fertilizer (33.3 kg/N/ha) was applied three times (June, July, and August) per year through the irrigation water during the fern growth period.

In late April or early May of 2016, 2017, and 2018, asparagus rooting depth and distribution were evaluated by soil coring [21,45]. Data were collected from four replications, three locations and six depths in each cultivar, and these data were used to determine fleshy root length, biomass, and root density. At random locations within the planting rows soil cores, 0.9 m long and 55 mm in diameter, were taken with a hydraulic soil corer (Giddings Machine Company Inc.; www.soilsample.com (accessed 6 December 2022)). Cores were collected at 0.0, 0.30, and 0.60 m from the row center. For samples collected at 0.0 m of the row center, the cores were taken between plants rather than through the crown. The 0.9 m extracted cores were then divided into 0.15 m intervals. In each year, the total number of samples collected was 216 (3 cultivars, 3 locations, 6 depths, and 4 replications). Fleshy storage roots were removed by hand from the soil in the field and stored in plastic bags. Fleshy roots were stored at 2 °C for up to 3 weeks [45]. Fleshy roots were washed, surface dried, and their length and diameter measured before drying at 80 °C for 48 h. Fleshy root length density (FL-RLD) was calculated as:

$$FL\text{-}RLD = L/V \; (m/m^3), \tag{1}$$

where L is the sum of the fleshy root length (m) and V is the volume (m$^3$) of soil core.

Asparagus cultivars were not harvested in 2016 to ensure that the plants had sufficient time to grow and develop the root system. The initial harvest began in 2017 (17–30 April: 10 cuts) and again in 2018 (24 April–23 May: 24 cuts). The number of harvests (cuts) did not match harvest intervals as seasonal weather conditions often limited spear growth and no harvests occurred on Sunday (harvest daily: Monday–Saturday). All spears taller than 23 cm were cut and graded (https://www.ams.usda.gov/grades-standards/asparagus-grades-and-standards, (accessed 22 December 2022)) into very large (+22 mm), large (18–22 mm), medium (13–18 mm), small (8–13 mm), very small (4–8 mm), or culls (<4 mm, bent/open heads, damaged, or unmarketable). On Tuesdays and Fridays, harvested spears were first weighed, then trimmed to 23 cm, graded for size, counted, and then re-weighed. Average spear weights by grades from those measured on Tuesday and Friday were then used to approximate spear weights on the day before or after to provide an estimate of productivity. Total productivity was the sum of the estimated and actual weights for the harvest period of each year. Ferns were evaluated in mid-August of each year. Random one meter sections of each row were assessed. Fern height was measured at three locations and total stem number per meter collected.

Root length densities, root dry weights, and spear and fern productivity were analyzed by standard analysis of variance to determine the main effects and interactions of year, cultivar, sampling depths, and location (where appropriate). The general linear model procedure (SAS Institute, Cary, NC, USA) was used for analysis of variance. Root length density distribution graphs were generated from these data using a contouring program in Surfer 13 (Golden Software, Inc. http://www.goldensoftware.com, (accessed 4 November 2022)).

## 3. Results

### 3.1. Fleshy Root Growth

Figure 1 illustrates the changes in fleshy root length density (FL-RLD; cm/cm$^3$) for three asparagus cultivars during the early establishment years of 2016–2018. For all the varieties evaluated, FL-RLD decreased with increasing depth in the soil profile and distance from the planted row regardless of year. The main effect of year was highly significant and FL-RLD increased with each year after planting. There was no interaction between year and cultivar ($p = 0.126$), therefore, each year was analyzed separately. In 2016, one year after planting, the majority of the FL-RLD was located near the planting row (0–30 cm distance) and more than 75% of the fleshy roots were found in the planting depth (0–30 cm) for each cultivar evaluated. By 2017, rooting depth increased and some fleshy roots were located as deep as 70–80 cm in the soil profile. By 2018, fleshy roots were found at the maximum sampling depth of 90 cm.

For the three asparagus varieties tested, while there were no significant differences in total FL-RLD, unique distribution patterns were noted (Figure 1). For the asparagus cultivar Atlas, FL-RLD was more uniform over depth and distance from the planted row. Isolines were widely spaced, indicating a more gradual change in root length throughout the soil profile. This pattern was evident in all three years and showed an increase in RLD as roots expanded outward and downward. For the cultivar Guelph Millennium, FL-RLD was concentrated in the upper 60 cm of the soil profile, isolines were closely spaced (indicating greater changes in FL-RLD with depth and distance), few roots grew below −70 cm in depth, and root growth was localized near the crown. These patterns were consistent from year-to-year as the root system grew. For the cultivar Jersey Giant, fleshy roots were initially concentrated in the upper 60 cm of the soil profile (2016), isolines were closely space (indicating large changes in FL-RLD with depth and distance), but in later years, root growth expanded outward and downward. These patterns were consistent from year-to-year as the root systems grew.

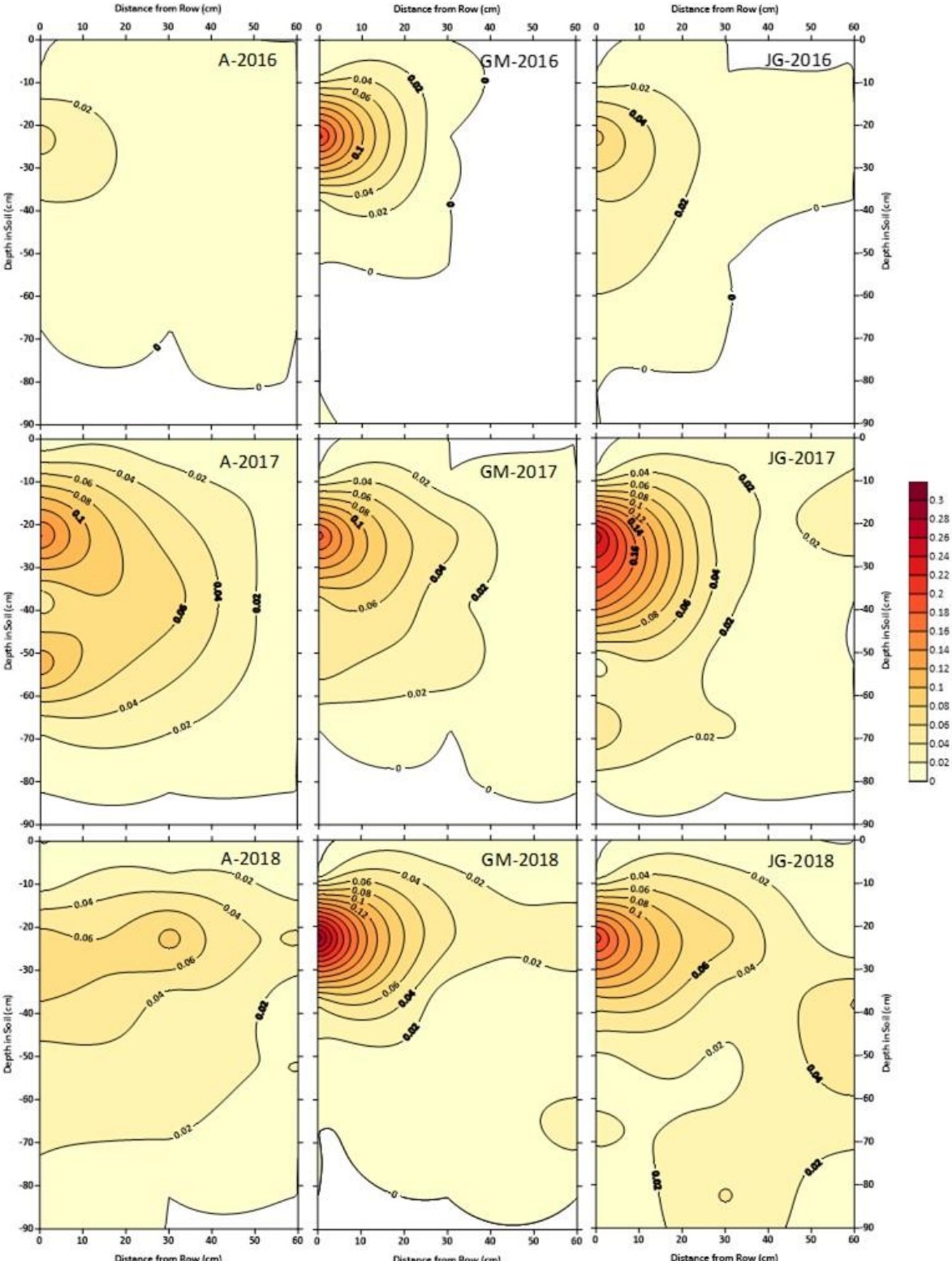

**Figure 1.** Fleshy root length density (FL-RLD) of three asparagus (*Asparagus officinalis* L.) cultivars (Atlas (A), Guelph Millennium (GM), Jersey Giant (JG)) for sampling years 2016, 2017, and 2018. The three cultivars (A, GM, JG) are aligned in the columns while the years (2016–2018) are aligned in the rows. Isolines represent the change in RLD (cm of root length per cm$^3$ of soil volume) over depth (0–90 cm) and distance (0–60 cm) from the row.

While the pattern of root distribution appears similar among the three asparagus varieties evaluated, there were notable differences (Table 1). In 2016, one year after planting, Guelph Millennium had significantly more fleshy roots (79%) located near the crown (0 cm distance; 0–30 cm depth) compared to Atlas (52%) or Jersey Giant (57%). In addition, there were significantly fewer fleshy roots below the crown (0 cm distance; 31–60 cm depth) for Guelph Millennium (5%) compared to Atlas (20%) or Jersey Giant (22%). A similar pattern was noted in 2017, two years after planting, where Guelph Millennium continued to have more fleshy roots located near the crown compared to Atlas or Jersey Giant. In addition, Guelph Millennium also had fewer fleshy roots below the crown compared to Jersey Giant, while Atlas was no longer different. By 2018, three years after planting, Guelph Millennium still had a higher percentage of fleshy roots in the shallow (0–30 cm) and intermediate (31–60 cm) depths of the soil profile compared to Atlas or Jersey Giant, though the differences were no longer significant. For both Atlas and Jersey Giant, fleshy roots also extended further into the interrow area (60 cm distance) and deeper (61–90 cm depth) into the soil than Guelph Millennium.

**Table 1.** Percentage of fleshy storage roots by depth and distance for three asparagus cultivars (Atlas, Guelph Millennium, Jersey Giant) during the early establishment years (2016–2018). Transplants planted in April 2015 and root growth assessed in May of 2016–2018.

| | Distance from the Row (cm) | | | | | | | | |
|---|---|---|---|---|---|---|---|---|---|
| | 0 | 30 | 60 | 0 | 30 | 60 | 0 | 30 | 60 |
| Depth (cm) | 2016 | | | 2017 | | | 2018 | | |
| | Atlas | | | | | | | | |
| 0–30 | 52 [B] | 15 | 5 | 39 [B] | 21 | 8 | 37 | 19 | 8 |
| 31–60 | 20 [a] | 5 | 2 | 17 [ab] | 9 | 2 | 10 | 16 | 4 |
| 61–90 | 1 | 0 | 0 | 4 | 1 | 1 | 4 | 4 | 1 |
| | Guelph Millennium | | | | | | | | |
| 0–30 | 79 [A] | 13 | 1 | 65 [A] | 10 | 3 | 44 | 25 | 4 |
| 31–60 | 5 [b] | 1 | 1 | 11 [b] | 10 | 1 | 12 | 8 | 3 |
| 61–90 | 1 | 0 | 0 | 0 | 1 | 1 | 2 | 0 | 2 |
| | Jersey Giant | | | | | | | | |
| 0–30 | 57 [B] | 10 | 7 | 41 [B] | 12 | 5 | 46 | 16 | 4 |
| 31–60 | 22 [a] | 3 | 0 | 22 [a] | 12 | 3 | 11 | 12 | 4 |
| 61–90 | 1 | 1 | 0 | 4 | 1 | 1 | 3 | 2 | 2 |

A,a,B,b: Depth difference. For each individual depth, least squares means with the same letter within a row indicates no statistical difference at $\alpha = 0.05$. If no letter is present, there is no statistical difference at $\alpha = 0.05$.

Estimated total fleshy root length and fresh weight per cubic meter of soil are reported in Table 2. There was no difference in total root length ($m/m^3$) between the three varieties evaluated in any year, however, years ($p = 0.018$) were significantly different from each other. In 2016, total fleshy root length averaged 67 $m/m^3$ across the varieties. Average fleshy root length increased to 146 $m/m^3$ in 2017 and was 361 $m/m^3$ in 2018. Total root fresh weight between the three varieties evaluated was not different in any year, but again years were significantly different ($p = 0.034$) from each other. In 2016, total fleshy root fresh weight averaged 0.58 $kg/m^3$ across the three varieties. This increased to 3.18 $kg/m^3$ in 2017 and was 4.67 $kg/m^3$ in 2018.

**Table 2.** Effect of asparagus (*Asparagus officinalis*) cultivars (Atlas, Guelph Millennium, Jersey Giant) on estimated total fleshy storage root length (m/m$^3$) and fresh weight (kg/m$^3$) during the establishment years. Transplants planted in April 2015 and root growth assessed in May of 2016–2018. (ns = not significant).

| Cultivar | 2016 | 2017 | 2018 |
|---|---|---|---|
| | Fleshy Root Length (m/m$^3$) | | |
| Atlas | 65.05 | 155.21 | 355.83 |
| Guelph Millennium | 52.58 | 182.40 | 374.15 |
| Jersey Giant | 82.77 | 201.98 | 374.51 |
| LSD 0.05 | ns | ns | ns |
| | Fleshy Root Fresh Weight (kg/m$^3$) | | |
| Atlas | 0.390 | 2.285 | 3.689 |
| Guelph Millennium | 0.609 | 3.828 | 5.530 |
| Jersey Giant | 0.739 | 3.428 | 4.790 |
| LSD 0.05 | ns | ns | ns |

*3.2. Crop Productivity*

Neither fern number nor fern height was influenced by asparagus cultivar ($p = 0.468$) or year ($p = 0.341$). Fern number ranged averaged 30–45 stems/m of row, and as plants grew older, fern number increased. Fern height varied from 0.65–0.7 m in 2016, increased to 0.90–0.95 m in 2017, and was 1.40–1.46 m in 2018. Height differences between varieties in all years were small and not significantly different.

Total marketable yield was not different between the three varieties in either 2017 (first harvest year) or 2018 (Table 3). Since the total number of cuts was different in each year, years were analyzed separately. In the short harvest of 2017, Atlas produced significantly larger diameter spears compared to Guelph Millennium or Jersey Giant. However, the yield of medium, small, and very small spears was not different between the varieties. As the asparagus plants grew, in the four-week harvest period of 2018, Atlas continued to produce significantly larger diameter spears compared to Guelph Millennium or Jersey Giant. However, the yield of large, medium, small, and very small spears was not different between the varieties.

**Table 3.** Effect of asparagus (*Asparagus officinalis*) cultivars (Atlas, Guelph Millennium, Jersey Giant) on spear productivity (kg/ha) during early establishment years (2016–2018). No harvest occurred in 2016, and the harvest years of 2017 and 2018 were analyzed separately due to differences in harvest lengths.

| Cultivar | 2017 Spear Yield (kg/ha) − (2–week harvest; 7 cuts) | | | | | |
|---|---|---|---|---|---|---|
| | Total Marketable | Very Large (+22 mm) | Large (18–22 mm) | Medium (13–18 mm) | Small (8–13 mm) | Very Small (3–8 mm) |
| Atlas | 1040 | - | 269 | 393 | 290 | 88 |
| Guelph Millennium | 937 | - | 187 | 373 | 290 | 86 |
| Jersey Giant | 857 | - | 149 | 321 | 288 | 100 |
| LSD 0.05 | ns | - | 57 | ns | ns | ns |

**Table 3.** *Cont.*

| | 2017 Spear Yield (kg/ha) − (4–week harvest; 24 cuts) | | | | | |
|---|---|---|---|---|---|---|
| Atlas | 2089 | 21 | 588 | 693 | 515 | 272 |
| Guelph Millennium | 1848 | 47 | 483 | 600 | 586 | 133 |
| Jersey Giant | 2228 | 73 | 503 | 697 | 746 | 209 |
| LSD 0.05 | ns | ns | 83 | ns | ns | ns |

## 4. Discussion

Past asparagus root studies focus on rooting depth, distribution, and root age in an attempt to identify field sites suitable for asparagus production [7,18–21]. Fleshy storage roots commonly grow 1–2 m in length, are believed to live for up to 6 years [7], can be altered by on-farm cultural practices [18,21,22,24,25], and grow best in lighter rather than heavier soil types. Less information is available on the dynamics of fleshy root growth in asparagus varieties as there are limited studies comparing them [24,42]. Our findings show that seasonal changes over several years results in downward and outward expansion of the root system. While total root length and root mass were not different between the three cultivars evaluated, there were differences in root distribution that may be important when making crop management decisions. Brainard et al. [26] speculated that there are differences in water-use between cultivars Jersey Supreme and Guelph Millennium. Jersey Supreme was better able to tolerate drought due to a deeper root system compared to Guelph Millennium. However, no root data was collected in their study. Asparagus cultivars Gijnlim and Guelph Millennium have different root distribution patterns in the first years after planting [42], and the Guelph Millennium pattern was similar to those reported in Figure 1.

Site selection for asparagus is based in part on planting in soils considered suitable for the particular needs of the crop [34–36,42]. Producers are advised to plant asparagus in free draining, deep, sandy soils and to avoid heavier silt or clay type soils, soils with water related issues, or those with known problems. Root distribution in asparagus changes with soil type and high yields are determined in part by rooting depth [18]. Liao et al. [46] reported higher spear yields and better spear quality when asparagus was grown in a sandy loam soil versus a heavier silt loam soil, but their study did not address root development issues. The better soil structure and depth of the Waimakariri (sandy loam) soil improved root distribution when compared to the Templeton (silt loam) soil [21]. While our study did not compare different soil types, rooting depth and distribution increased every year and patterns looked similar to those previously reported [21,42]. Cultivar difference can also be important when selecting sites to grow asparagus [20,34–36]. Our findings show that rooting depth and distribution were not negatively impacted by the silt loam soil used in this study, but cultivars have quite different root distribution patterns, which could be important. Cultivars with roots concentrated in less soil, such as Guelph Millennium, may be more susceptible to biotic stresses particularly early in the production cycle. Having more detailed root distribution information along with soil series may help growers identify key production problems and provide insights about how to improve asparagus growth, management, and productivity.

While there were no significant differences in total productivity between the three cultivars evaluated (Table 3), Atlas did produce more large spears than Jersey Giant and Guelph Millennium. Atlas is known to produce larger diameter spears. It has been reported that asparagus productivity is related to root carbohydrate storage [8–14] and root size can influence total CHO storage [21,42]. The size of the root system differs among cultivars, depends on plant population, is very dependent on soil conditions, changes each year, and increases during the first few years as crops are establishing [14]. In established mature asparagus fields in New Zealand, root mass ranges from around 0.12 to 1.2 kg

dry weight/plant (approx. 1.0–10.5 kg fresh weight/m$^3$). While carbohydrates were not measured in this study, root biomass values (Table 2) are within the range described [14] for a young asparagus field. Additional root growth is expected and since yields and biomass values were similar for the three cultivars, one may assume carbohydrate levels were not different. Regular assessments of root growth in mature asparagus fields may not be necessary because changes in structural root biomass is slow in fully established fields [14]. However, in young developing fields, root growth variations may contribute to yield variability, thus mapping root development may identify differences due to cultivars, management, or other abiotic or biotic factors.

Asparagus growers regularly make important crop management decisions that can impact asparagus productivity. These include decisions related to compaction, weed management, irrigation, and nutrient management. Tillage operations in asparagus loosen the soil before spear emergence, build soil ridges to improve spear size, and/or incorporate fern residues, and control weeds [22,23,40]. Tillage has been shown to reduce long-term asparagus performance [22,23]. During the establishment years, tillage operations fill in the planting furrow and reduce annual weed pressure. These tillage operations significantly reduced root growth near the soil surface [24]. Winged tines of different configurations were very effective in alleviating compacted soil regions [40,41], however, proper equipment is required to avoid crown and surface root injury during bed formation and weed control. Given the differences in root distribution between the three asparagus cultivars tested, growers using Guelph Millennium should carefully monitor tillage to minimize possible root damage [39,42]. Tillage damage to crowns and the roots near the crown ultimately reduce fleshy root extension and fibrous root growth [16,18]. Other studies indicate that new fleshy roots are formed after harvest [19,24], therefore, tilling after-harvest often slows fern establishment [22,23] and further interrupts crown expansion. Consequently, tillage is commonly responsible for shortening the life span of asparagus fields and for lowering average yields.

Other studies on asparagus have shown that water stress in one year can decrease productivity in the following year, with the decrease in productivity due to a decrease in spear size and weight but not spear number [25,27,43]. Brainard et al. [26] speculated that there are differences in water-use between cultivars Jersey Supreme and Guelph Millennium. Jersey Supreme was better able to tolerate drought due to a deeper root system than Guelph Millennium. While water use was not part of our study, our root distribution pattern differences between three asparagus cultivars would suggest differences in irrigation management may be necessary depending on cultivar selected. Earlier studies by Drost and Wilson [21] and Drost and Wilcox-Lee [24] only evaluated root growth changes over one production season and limited their efforts to one asparagus cultivar. Varietal differences in water use, based on soil water differences, imply differences in rooting patterns [26]. The findings reported here illustrated that Guelph Millennium roots were more horizontal (shallow roots) and concentrated in the upper 50–60 cm of the soil profile in the early years after planting. In contrast, Jersey Giant (a closely related cultivar to Jersey Supreme) and Atlas both have roots systems that extend deeper into the soil profile (Figure 1; Table 1). These distribution differences may explain those water use differences noted by Brainard et al. [26] and careful selection of varieties could improve drought tolerance. Schaller and Paschold [44] endorsed the hypothesis that there are asparagus genotype specific drought vulnerabilities that are based on stomatal characteristics. While there was no mention of the root development characteristics of the two varieties evaluated, one may speculate that there are also differences in rooting patterns that will require additional study. Growers, crop advisors, farm managers, and researchers should pay more attention to crop rooting patterns. Root biomass and distribution information along with the root carbohydrate content could provide growers with detailed information regarding asparagus's health and productivity potential [21]. Root samples collected over many years reveal changes in root development that may provide clues to asparagus decline, variations in performance,

and weaknesses in production practices. This type of information can be used to improve management and increase seasonal productivity and field longevity.

Fleshy roots emerge from the crown near the growing apices and grow for several years [7,15,19]. Therefore, fleshy root development patterns should look the same throughout the season, or from year-to-year, which is evident in this study. The year-to-year differences in rooting pattern noted in Figure 1, while showing similar appearances, also reflect the active growth change that occurs over time. One year after planting (2016), fleshy roots extended down to 60 cm deep and out 60 cm from the plant. By Year 3 (2018), fleshy roots extended past our sampling depth and out beyond the middle of the rows. Other work by this author illustrated that there were few in season changes in fleshy root growth in mature asparagus [21,23,24]. Given that fleshy root growth changes slowly, periodic sampling during the early years after planting is a simple way to assess root growth changes, and it provides an accurate way to evaluate asparagus growth, identify production weaknesses, and highlight management practices that may hinder productivity. Further root sampling of important asparagus cultivars grown world-wide could provide us with a better understanding of asparagus growth and the variables that impact crop yield.

## 5. Conclusions

Several conclusions can be drawn from these studies. Firstly, novel procedures are needed to collect roots since they are difficult to access. Simplified sampling techniques are needed to assess rooting depth and distribution. The technique used required minimal time and manpower, and adequately illustrated the differences in rooting patterns of asparagus cultivars. Secondly, the patterns noted from this study suggest that crop management strategies (nutrient and water application practices) should be tailored to the specific growth patterns of the cultivars grown. This would ensure that plant stress is minimized. Growers and managers of asparagus need to realize that management practices may adversely affect field longevity and seasonal productivity. Thirdly, knowledge of the size of the root system is needed to estimate if there are adequate carbohydrates to fuel high productivity and maintain longevity in asparagus. Thus a measure of total root biomass is needed in addition to carbohydrate content to ensure crop health and yield potential. Finally, additional work on root development changes over time is required for the important asparagus cultivars grown. This additional information will help improve our understanding of the yield physiology of asparagus.

**Funding:** This research was funded (UAES1685) by the Utah Agricultural Experiment Station, 4800 Old Main Hill, Utah State University, Logan, UT 84322-4800; journal paper number UAES #9634.

**Data Availability Statement:** Data are contained within the article.

**Acknowledgments:** The assistance with data collection and processing of James Frisby of Utah State University and our undergraduate students, Josh Martin, Andrew Bohannon, Emmalee Rolfe, and Evan Christensen was greatly appreciated.

**Conflicts of Interest:** The author declares no conflict of interest. The funders had no role in the design of the study; in the collection, analyses, or interpretation of data; in the writing of the manuscript; or in the decision to publish the results.

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
