# Peer review of "Asparagus (Asparagus officinalis L.) Root Distribution Varies with Cultivar during Early Establishment Years"

_horticulturae, doi:10.3390/horticulturae9020125_

Round 1
Reviewer 1 Report
Understanding root development in any cultivated species is critical to achieving high productivity. In asparagus, this importance is even greater, due to the perennial nature of the plant and the crop, but also the dependence of productivity on the levels of stored carbohydrates (CHO) in the root system. Even though previous research showed differences between cultivars in the utilization of water and nutrients, but also yield potential etc. and implied differences in the development of the root system, available information on varietal differences in root growth patterns is non-existent. The submitted manuscript describes the changes in root growth of three asparagus cultivars over the three growing seasons. The results revealed differences in root distribution patterns between cultivars (although there were no differences in root length or fresh weight), as well as in root growth over time.
Of the results, notable is (Lines 222-227): In both harvest periods (2017 and 2018), ‘Atlas’ produced significantly more large diameter spears compared to ‘Guelph Millennium’ or ‘Jersey Giant’. Certainly, further comment and discussion of these results considering the differences in root distribution is required, which the author can provide from his own experience, given that there is apparently no such thing in the relevant literature.
Author Response
I have integrated into the discussion additional thoughts on root growth and discussed the yield differences noted.
Reviewer 2 Report
Article well done
Author Response
Thank you for your kind words, see additional details and changes as per other reviewers.
Reviewer 3 Report
In this work, Drost performed the changes in root growth that occur over the three establishment seasons of three asparagus cultivars. The author collected root growth data from 2016 to 2018, and selected four replications, three locations and six depths in each cultivar per year. Then one-way analysis of variance and descriptive analysis were performed on the data.
Line 8-9, the work is mainly a comparison of different cultivars, soil type,crop management practices and other factors are not reflected.
Line 102-104, what are the characteristics of these three cultivars and why are these three cultivars selected for research?
Line 220-221, how is the total marketable measured, is it the same set of data as mentioned in line 141-142?
Line 329-330,there were no variables involved in crop management strategies in the study.
Author Response
Line 8-9, True; Most studies looked at one season of root growth and have not evaluated changes over time (published literature). Our work builds on this and describes cultivar differences.
Line 102-104, Added details on each cultivar into the M&M
Line 220-221, Additional details included in the M&M.
Line 329-330, True, it is hard to manage a crop without knowledge of root distribution or mass. Practitioners of asparagus need to gather root information in addition to above-ground growth parameters if they are going to manage the crop successfully.
Reviewer 4 Report
It is an intriguing paper on a current topic; I appreciate the quality of the experiment and the highly-developed discussion. I have no reservations, except for the following comments:
Better define scientific hypotheses
Tables 1 – 3: edit the format
What is the effect of applied herbicides on asparagus?
Author Response
Thank you, see additional details and changes as per other reviewers. Also, I altered the hypothesis as suggested; reformatted the three tables. We saw no effect of applied herbicides. Since this was not the focus of the project and we have been using this combination in other studies over many years, we do not believe it influenced our results.
Round 2
Reviewer 3 Report
In this work, Drost performed the changes in root growth that occur over the three establishment seasons of three asparagus cultivars. The author collected root growth data from 2016 to 2018, and selected four replications, three locations and six depths in each cultivar per year. Then one-way analysis of variance and descriptive analysis were performed on the data.